# Multifaceted Communication Problems in Everyday Conversations Involving People with Parkinson’s Disease

**DOI:** 10.3390/brainsci7100123

**Published:** 2017-09-25

**Authors:** Charlotta Saldert, Malin Bauer

**Affiliations:** Institute of Neuroscience and Physiology, Speech and Language Pathology Unit, University of Gothenburg, Gothenburg SE-405 30, Sweden; malin.bauer@neuro.gu.se

**Keywords:** Parkinson’s disease, communication disorder, motor speech disorder, dysarthria, anomia, conversational interaction, spouses, Conversation Analysis

## Abstract

It is known that Parkinson’s disease is often accompanied by a motor speech disorder, which results in impaired communication. However, people with Parkinson’s disease may also have impaired word retrieval (anomia) and other communicative problems, which have a negative impact on their ability to participate in conversations with family as well as healthcare staff. The aim of the present study was to explore effects of impaired speech and language on communication and how this is managed by people with Parkinson’s disease and their spouses. Using a qualitative method based on Conversation Analysis, in-depth analyses were performed on natural conversational interaction in five dyads including elderly men who were at different stages of Parkinson’s disease. The findings showed that the motor speech disorder in combination with word retrieval difficulties and adaptations, such as using communication strategies, may result in atypical utterances that are difficult for communication partners to understand. The coexistence of several communication problems compounds the difficulties faced in conversations and individuals with Parkinson’s disease are often dependent on cooperation with their communication partner to make themselves understood.

## 1. Introduction

Clinical-perceptual analysis have revealed that up to 89% of people diagnosed with Parkinson’s disease (PD) also have a progressive motor speech disorder, usually a hypokinetic dysarthria. This is characterised by a monotone and breathy voice and imprecise articulation [1]. It may severely impair speech and prevent affected individuals from being able to make themselves understood. In particular, voice problems, such as hypophonia, make it difficult for the person with PD to join in and be heard in conversation. Social interaction and daily life thus become challenging [2,3]. 

Effects of PD on cognition have also been noted [4,5]. Dementia is strongly associated with PD, but people who have PD may experience cognitive impairment without being diagnosed with dementia [6]. Studies have shown that memory, visuospatial abilities, executive functions and working memory are among the first cognitive functions to be affected [7]. Language deficits that develop through the course of the disease have also been reported and these affect not only comprehension but also production [7,8,9,10]. This impairment results in less informative speech content and effects on performance on verbal fluency tasks have also been demonstrated [11]. In interviews, people at various stages of PD describe that they are experiencing word retrieval deficits (anomia), problems with sentence formulation and impaired everyday language comprehension [3]. These language deficiencies appear in people who do not meet the criteria for dementia. 

The findings of language and cognitive deficiencies have guided researchers to examine so-called *pragmatic* impairments associated with PD [12,13]. Word retrieval difficulties are usually regarded as affecting the semantic aspects of language. However, pragmatic ability may be defined as the ability to use linguistic and non-linguistic communication skills in particular contexts [14]. Linguistic skills include comprehension and the use of correct syntax and appropriate semantics, while non-linguistic skills refers to the appropriate use of such things as gestures, gaze and voice modulation. Pragmatic impairment impacts on an individual’s ability to effectively communicate their desires and opinions [13]. Formal tests have been used to examine the pragmatic ability of individuals with PD [12,13,15,16,17], and some studies have also analyzed conversational interaction [2,13,18,19].

People tend to adopt certain communication behaviors, or patterns, such as patterns for turn-taking to facilitate mutual understanding [20,21]. The term *repair* refers to the way in which conversation partners highlight and manage a situation in which mutual understanding is at risk [22]. It may be that the speaker has difficulty finding words or fails to give sufficient information, or the listener may have problems understanding because of impaired hearing or because the speaker articulates poorly or quietly. Generally, the speaker will either try to repair immediately or their listener may do so at their next turn [22]. However, if one of the communication partners has a communication disorder, the repair process may take several turns and may require cooperation between the participants in the conversation [23,24,25].

It has also been noted that people without communication disorders tend to use certain structures for topic manipulation [26,27]. Still, they too sometimes find it difficult to follow shifts in topic and mutual understanding may then suffer and need to be repaired [28]. Although casual conversation often involves a flow from one topic to another without explicit introduction, the conversers often collaborate in creating topical coherence [29]. In general, people prefer some form of linking transitions. Transitioning is usually achieved collaboratively, with the conversers using techniques such as pausing to signal that they have agreed to discontinue a topic. They may then open a new, related topic. Alternatively, they may introduce an unrelated topic using a phrase such as “by the way” or some other explicit marker. When one of the participants has a motor speech disorder, shifts in topic are often problematic [30]. It is difficult to interpret the impaired speech without knowing what the theme or topic is.

Perkins [14] notes that pragmatic ability is the individual’s capacity to adapt and make use of available verbal and non-verbal resources to communicate effectively in a given situation. This means that behavior registered in a test or in conversation as a symptom of impairment may actually be a strategy the person uses to compensate for the impairment and to facilitate communication. This is in line with a study by Illes et al. [31] in which the authors argue that the changes in syntactic production seen in people with PD may in fact be an adaption to their speech impairment. Reduced syntactic complexity and less frequent use of function words and interjections may occur because the person is excluding less important words and focusing on content words instead to get their message across.

Studies of conversations in which one of the partners is diagnosed with PD have revealed that they often have problems initiating speech and their speech may often occur in overlap with their communication partner’s speech. In some cases, this may give rise to long repair sequences with repeated attempts to initiate and complete repair [2,18]. Others have observed word retrieval difficulties including the use of unspecific vocabulary and atypical use of words [19]. 

Research has so far predominantly focused on the effects of either speech impairment or word retrieval problems and there is now a call for appreciation of the multifaceted effects of PD on communication [32]. The aim of this study is therefore to explore how different types of communicative problems may compound one another in people diagnosed with PD and to see how they and their spouses manage this in everyday conversations.

## 2. Materials and Methods

Five men diagnosed with idiopathic PD (PwPD) and their spouses (CP) were recruited from the local PD association. All the PwPDs had a motor speech disorder, diagnosed as hypokinetic dysarthria and they reported that they had trouble making themselves understood in everyday conversations. Word-fluency was explored with phonological and semantic verbal fluency tasks [33], comprehension with the Token test [34], and cognition with the Mini Mental State Examination (MMSE) [35], see Table 1. The degree of comprehensibility in the PwPDs’ speech was measured using a ten-minute audio recording of a conversation between the two spouses and then calculating the percentage of words correctly understood by a naive assessor [36].

All participants had Swedish as their mother tongue. Exclusion criteria for all participants were: (other) neurological disease or injury; drug or substance abuse; vision or hearing impairments that were not compensated for with aids. None of the participants had been diagnosed with dementia. However, the results on language and cognitive screening tests indicated cognitive decline in three of the PwPDs (Dyads 1, 3 and 4). 

The material consisted of 255 min of everyday conversations from 17 video recordings. These were each 15 min long and they were recorded in the participants’ homes. The amount of data differs between the dyads since Dyad 4 and Dyad 5 also participated in another study in which more data was collected. Since the five dyads were not being compared with each other and in order to increase the validity of the study, we included all available data in our analysis. The dyads were instructed to interact as they normally did and they were told that they did not have to talk all the time but could remain silent if they wished. Two cameras were set up to capture both partners’ use of non-verbal communication. A separate audio recording was made to capture more of the dysarthric speech. The dyads were left alone while the recordings took place. The study was approved by the regional ethical review board (Gothenburg: 102-10, 29 August 2016) and all names have been changed in order to protect the participants’ identity. 

The analysis was based on Conversation Analysis (CA) [38,39]. CA, which is rooted in ethnomethodology, is an inductive, data-driven qualitative method used to study conversation in a natural setting. In CA, human interaction is viewed as structured and orderly, following conventional patterns in a sequential context. Each observed instance of a repair sequence was transcribed and further analyzed. Repair initiated by the conversation partner (CP) was defined as an instance in which they signaled, verbally or non-verbally, that they had not understood the previous utterance. For those diagnosed with Parkinson’s disease (PwPD), instances of repair included clear signs of hesitation, pauses, false starts or repetition. 

Transcription conventions based on CA guidelines [40] were used (see Appendix A for a key to transcription symbols). The transcribed extracts presented here have been translated from Swedish to English. 

In accordance with CA methodology, several data sessions took place to ensure inter-reliability and validity of the analysis. In these sessions, the second author presented transcripts and video extracts to researchers who were familiar with CA methodology and communication in PD, but they were not aware of the aims and purpose of the study or any preliminary results. The findings were discussed during these sessions and consensus was achieved on analyses that deviated from the preliminary results. 

## 3. Results

Two hundred and twenty-one initiations of repair were observed, and these were reasonably equally distributed between the five dyads (see Table 2). 

In-depth analysis of the video recordings revealed that in addition to the problems associated with impaired speech, there were problems of a semantic and pragmatic nature. Three broad but distinct types of problem were identified. These related to: (1) impaired speech; (2) word retrieval problems; and (3) pragmatic problems, though these are not mutually exclusive categories. The instances of repair were not registered under just one of these three types since they actually represent different aspects of the communication difficulties observed. In other words, the different types of problem often occurred concurrently. For example, an utterance that includes a paraphasic word is harder to understand if there is also motor speech disorder. Similarly, when there is a motor speech problem, this makes it harder to follow a topic shift, particularly if the shift is not clearly demarcated. Further, the strategies used to handle word retrieval problems or impaired speech may result in a pragmatic problem.

The results on the measure of degree of comprehensibility, presented in Table 1, showed that all the participants had dysarthria (mean = 41.4%, range: 23–83%). Impaired speech was also the most apparent problem in the majority of the repair sequences. The PwPD-5 had the lowest degree of comprehensibility (23%) and the dyad also had the most frequent occurrences of repair, see Table 1 and Table 2. Of 221 observed repair initiations, approximately two thirds were caused by a speech problem. These repair sequences were often initiated by the CPs with a request for clarification and were then resolved by the PwPD repeating parts of their utterance, sometimes in a louder or clearer voice. When speech was severely dysarthric, the CP often had to repeat the request for clarification and the PwPD had to make several attempts at repair before the problem was solved. However, the relation between degree of comprehensibility and frequency of repair in the dyads was not direct. Although dyad 3 had as many occurrences of repair as dyad 4, the degree of comprehensibility for PwPD-3 was as high as 83% compared to only 31% for PwPD-4. In about a third of the repair sequences in the entire data, problems of a semantic and pragmatic nature compounded the difficulty in achieving mutual understanding. Five transcribed extracts from the data are presented below to exemplify the three types of problem. 

### 3.1. Impaired Speech Signal

The first extract is taken from of a conversation between the spouses in Dyad 1. During the conversations, PwPD-1 sits in a slouched position, makes little eye contact and uses few facial expressions or gestures. His spouse reported that he had normal hearing, but his cognition was severely impaired and he was not able to participate in the Mini-Mental state examination [35] or the formal test of language comprehension [34]. In the conversational interaction he shows obvious examples of impaired language comprehension. His articulation is imprecise and his speech is very quiet. His spouse often finds it hard to understand what he says and she sometimes initiates repair but sometimes just abandons the topic.

This conversation was recorded in winter and the couple was talking about the weather and discussing whether or not it was going to snow, see Figure 1.

The dyads main objective here is to maintain friendly interaction. Because of PwPD-1’s cognitive decline, CP-1 prioritizes social interaction over the exchange of accurate information in all the video-recordings. The way the repair is performed in this extract indicates that the source of the problem is impaired speech.

On line 02, PwPD-1 agrees with CP-1’s suggestion that snow will come. He then initiates an elaboration of the topic and overlaps with CP-1’s speech on line 04. On line 05, he lets CP-1 know that he is quite confident that snow will eventually come. However, his motor speech disorder makes his voice very weak here. His spouse leans towards him and initiates repair (line 06) by asking him what he said. He quickly resolves this by repeating his utterance, now with a little less certainty (line 07), and his wife confirms that she has understood by repeating his words (line 08). Further evidence of the fact that the problem was related to his impaired speech is that his wife (line 08) leans back again to her original posture, thus indicating that repair was successful. 

Although the problem in this case was quickly resolved, in many cases it took several requests for clarification, some guesswork and repetitions before CPs understood, (see Figure 2 below from the conversations in Dyad 5). 

PwPD-5 has severe motor disabilities. Consequently, CP-5 has great difficulty understanding him and she constantly repeats what she thinks he is trying to say to make sure she has understood. When PwPD-5 initiates new topics or elaborates on a topic and CP-5 finds it hard to follow, either she or PwPD-5 initiates repair. In this extract, they are sitting on the sofa in their living room. PwPD-5 is more or less lying down and he has his feet in CP-5’s lap. They have been talking about how CP-5’s work is going and then PwPD-5 elaborates by asking how a common acquaintance is doing. PwPD-5’s articulation is severely impaired and his voice is breathy.

CP-5 asks for clarification twice (lines 02 and 04) and PwPD-5 repeats the utterance in the same simplified form three times (lines 03, 05 and 07). On line 08, CP-5 finally provides a suggestion that PwPD-5 accepts.

### 3.2. Word Retrieval Problems

In the extract in Figure 3, the spouses in Dyad 3 are sitting in their kitchen, sharing some memories. PwPD-3 has mildly dysarthric speech, difficulty initiating speech and a soft voice. His speech is quite comprehensible (83%) but he frequently shows signs of anomia and mixes up words (semantic paraphasias). PwPD-3 often recognizes his mistakes and initiates repair himself, usually signaled by pauses and hesitation. He tends to look to his spouse for help in these situations. She then usually offers alternatives or provides him with the first part of the word she thinks he is looking for. In the extract, they have just ascertained that except for the period of PwPD-3’s military service, they have hardly been apart since they first met.

In this extract, Dyad 3 performs the repair and reaches mutual understanding quite quickly. On lines 01–02, PwPD-3 is talking about how they have even been working at the same place. CP-3 agrees on line 03 and comments on how much time they have shared together (lines 04–08). PwPD-3 produces an inaudible utterance that overlaps with CP-4’s speech on line 07, and then on line 09, PwPD-3 completes CP-4’s sentence by saying “known each other”, which is pragmatically somewhat out of place here. PwPD-3 immediately initiates a repair on line 09 by using the cut-off disjunctive “or” and three more syllables that cannot be interpreted from the recording. 

The expression used here may be considered an unintended mix up of words—a semantic paraphasia. On line 10, CP-3 completes the repair by suggesting the more appropriate expression “been together” to capture the fact that they have been in each other’s company almost constantly since they got married. The PwPD-3 accepts and confirms CP-3’s suggestion on line 11. Nothing in this extract indicates that CP-3 is having difficulty interpreting the speech signals. Instead, the couple deals with the source of the trouble and the CP can complete the repair (line 10) by exchanging the phrase to a pragmatically more suitable expression before they move on. 

The next extract, in Figure 4, also illustrates word retrieval problems. It is taken from a conversation in which Dyad 4 is sitting at the kitchen table looking at a photo album. Although CP-4 tends to dominate the conversation, she often makes room for PwPD-4 and invites his participation. PwPD-4 tries to contribute but rarely takes the initiative. When he does contribute, his speech is often difficult to understand because of his indistinct articulation, hypernasal voice quality and signs of anomia. PwPD-4’s degree of comprehensibility in conversation was 31%, see Table 1. His spouse frequently initiates repair by asking him to speak up, and by repeating and making suggestions as to what she thinks he is trying to say.

In the extract in Figure 4, the couple is looking at pictures of their farming machines and PwPD-4 is having difficulty explaining to CP-4 why he is unhappy about a picture of a combine harvester. This problem is more complicated than that in Figure 3, and repair takes a little longer. On line 02, PwPD-4 points to the picture and says that he wants to “put in a photo of” (it) with indistinct articulation. CP-4 then initiates repair by trying to repeat what she thinks he said on line 03 and emphasizing the word “in”. However, while PwPD-4 accepts her interpretation after a short pause (line 04), on line 05, CP-4 attends to a problem with the content of what PwPD-4 has said, since the photo has already been put *in* the album.

PwPD-4’s utterance on line 02 is difficult to understand in this context and there is therefore a pragmatic problem with it. CP-4 continues on lines 07–08 by asking if he wants a picture of something else, but PwPD-4’s inaudible response (on line 09) results in another request for repair by CP-4. On lines 09–13, PwPD-4 again tries to explain what he means. His speech is severely dysarthric here and on line 10, CP-4 requests clarification after PwPD-4 has had difficulties to start his speech, in line 09. Then, on line 12, she seems to be able to interpret what he says because she asks supplementary questions. She argues that he already has the photos he is asking for. The problem is not solved until line 15, where PwPD-4 manages to explain that he wants the pictures to be bigger. CP-4 finally understands what he means and provides him with the missing word “enlarge” (line 16), and the couple can then continue looking through the album. 

There are a number of factors that contribute to this rather long repair sequence. Firstly, PwPD-4’s utterance “put in a photo” follows a one-second long pause (line 02), which indicates that he is having problems finding the words he needs to express what he wants to say. The expression “put in photo” should not be regarded as paraphasia, or an unintended mix up of words. It may instead be a first, intentional or unintentional, attempt to find a way of conveying that he wants to have an enlarged photo put in the album when he is unable to find the correct word. That is, to make a circumlocution. It is impossible to say whether this is a conscious strategy or an automatic adaptation to his word retrieval difficulties, but whichever it is, it is insufficient here and results in a pragmatic problem. His wife is unable to understand him and he relies on her to repair here. When he responds to CP-4’s questions, he is finally able to retrieve the word *larger*, and this then helps CP-4 understand him and complete the repair. However, there is not only a word retrieval problem and a circumlocution strategy here, but the husband’s motor speech disorder also creates difficulties. These problems compound one another such that it takes time to sort out the communication. 

### 3.3. Pragmatic Problem Due to Topic Shift and Timing of Information

The example in Figure 4 above highlights an important feature of communication problems in PD that was noted throughout the data. This is that several sources of difficulty tend to co-exist when people with PD have trouble making themselves understood. In the next extract, a topic shift seems to be the main problem, but this is compounded by dysarthric speech. 

The extract in Figure 5 is drawn from a conversation held by Dyad 2. PwPD-2 is not noticeably cognitively affected by his disease, and only a few word retrieval problems occur in his conversations. However, his facial expressions, posture, and speed and range of movements are clearly limited. His articulation is imprecise and his voice is very soft and leaky. CP-2’s main strategy for dealing with this is to ask him to clarify and to make suggestions for what she believes he just said. 

In the extract in Figure 5, the couple is sitting by the coffee table in their living room. The main topic of their conversation has been how to build a steel frame over PwPD-2’s bed so that he will be able to get in and out of the bed and control the TV by himself. The extract starts with CP-2 abandoning this main topic to comment on their cat, which is lying down by one of the video-camera bags. PwPD-2 does not contribute to the new topic introduced by CP-2 in line 01 and the two pauses that follow (line 02 and 04) indicate that the topic is about to be closed, although CP-2 elaborates on it on line 03. After the two-second pause on line 04, the topic seems to have been exhausted and she looks at her watch. On line 06, PwPD-2 introduces a new topic by saying: “the antenna”. On line 07, CP-2 initiates a repair by asking for clarification and PwPD-2 elaborates on the topic of the antenna (line 08). On line 09, it seems CP-2 believes she has understood and asks a question in line with her interpretation of what her husband said. However, it is in fact not at all related to what he said. He rejects her interpretation and repeats the word “the antenna” (line 10). When CP-2 confirms the correct word (line 11), the topic is established and the conversation can move on. 

Following topic shifts can be challenging for anyone but they are particularly problematic in the presence of dysarthric speech [28,29,30]. It is often recommended to people with communication disorders that they present the topic first in an utterance to make it easier for their partners to understand [41]. PwPD-2 may therefore have mentioned the antenna—the new topic of conversation—as a communicative strategy to help CP-2 understand.

However, the extract in Figure 5 also illustrates a pragmatic problem in the way PwPD-2 uses language. Although PwPD-2 introduces a new topic by mentioning “the antenna”, this alone does not provide sufficient information. There is a problem with the timing and PwPD-2 gives no transition markers that relate this topic to that of the steel frame that they had been talking about before (like for example: “talking about the frame we’re building for the TV, I just thought of the antenna”). Nor is PwPD-2 able to give any further clues until CP-2 initiates repair. He does not elaborate on the new topic until line 08, and again once the topic has been established, on line 13. The sparse information that he gives may be another strategy, an adaptation to his motor speech disorder, with which he gives less information in each utterance in order to maximize the clarity of each word [31]. However, this together with the shift in topic makes it difficult for CP-2 to follow. 

## 4. Discussion

This study supports the observation that the sources of communication problems in PD may involve not only a motor speech disorder but also semantic and pragmatic issues [12,13,19]. The qualitative analysis performed here reveals that people diagnosed with PD may not only have impaired speech in conversations but may also have difficulty retrieving words. This means they may fail to communicate essential information, mix up words or use atypical words and phrases. If they have problems retrieving words, they may also try to circumvent the problem and all this can lead to further pragmatic problems, which compound the impaired speech and make it even more difficult for these persons to make themselves understood. The study also revealed that utterances that may be considered to be atypical use of language, and thus as a pragmatic problem, may instead be regarded as the PwPDs’ strategies to adapt the way they speak to their disability. This means that something like a phrase used out of context (Dyad 2: “this I want (1.0) to put in a photo of”) or a poorly introduced new topic (Dyad 4: “the antenna”) may indicate communicative competence. This way of viewing pragmatic ability in general has been highlighted by Perkins [14], and the perspective has been discussed in relation to people with PD using simplified syntax [31]. The phenomenon has also been described in relation to aphasia [42,43] and other conditions that are associated with communication disorders, such as Huntington’s disease, amyotrophic lateral sclerosis and autism [44,45,46].

Circumlocution in cases of anomia and presenting the topic first when introducing a new topic can be effective strategies for people with communication disorders [41]. This means that although something that is said strategically may initially seems to create more problems, it may actually be helpful in the end. However, the data presented here, in likeness to that discussed by Saldert et al. [19], shows the importance of cooperation and support from the communication partner. It has been demonstrated that CPs of both people with PD and people with aphasia use similar communicative strategies when having difficulties understanding their partner [47]. Individual characteristics in terms of degree of communication disorder as well as the communicative style and attitudes of persons with communication disorder and their CPs seem to be more decisive for how the communication problems are handled. Further, the CPs attitude and communicative style is usually expected to be related to their knowledge about and understanding of the communication disorder [48,49].

As Perkins [14] argued, factors both within the individual and in their environment, such as their partner, affect a person’s ability to communicate. In PD, poor articulation and voice quality also interact with word retrieval problems and sometimes also inefficient communication strategies and adaptations. Further, their general motor disability makes it difficult for a person with PD to use non-verbal strategies such as facial expressions and gestures to support what they say, and cognitive deficits, such as impaired executive functions, may also impede them [8]. In combination, these factors may compound one another, and result in a negative synergistic effect. In contrast to Saldert et al.’s findings [19], most of the incidences of repair initiation in this study could be related to the impaired speech itself. The lower frequency of semantic or pragmatic sources of difficulty may be explained by the fact that the participants in this study had more severe dysarthria than those of the previous study. In this study, four of the five participants had moderate to severe dysarthria and their spouses dominated conversation and took most of the responsibility for repairing it. The other participant (PwPD-3) had 83% comprehensibility and was also the one who demonstrated the most obvious word retrieval problems. However, it should also be noted that severely dysarthric speech is difficult to transcribe verbatim, so it is possible that the incomprehensibility of their speech was masking instances of word retrieval problems and circumlocutions. A lack of evidence of semantic-pragmatic sources of problems for people with more severely impaired speech may also be accounted for by the fact that the spouses tended to do most of the talking, leaving less room for the PwPDs to speak. Also, the partners may refrain from initiating repair because they know how hard it will be to complete it because of the dysarthric speech. It seems to be relatively common for spouses to people with PD and also spouses to people who have severe aphasia after a stroke to avoid trying to repair communication [47]. This means that it cannot be inferred that interactions with few occurrences of repair are always functional, at least not if the transmission of information is more important than the social interaction in the conversation.

Impaired hearing is common in older age [50], and particular in relation to PD [51]. Impaired hearing may of course have impacts on the need for repair in conversational interaction and was an exclusion criterion for both the PwPDs and the CPs in this study. A limitation of this study is that the hearing was not formally tested, only self and other reported, but the difficulties experienced by the researchers in transcribing the speech that preceded repair indicates that the trouble cannot be attributed to hearing loss among the CPs. 

The examples presented here illustrate how couples manage to repair communication problems cooperatively. Previously, this process has been described mainly in relation to people with aphasia [25,52], but more recently also in those with PD [2,18,19]. The CPs try to help the PwPDs repair the interaction as quickly as possible and the PwPDs depend on their CP’s support to make themselves understood. As the behavior of the CP may both impede and facilitate the communication this adds to the compound relationships between factors such as degree of dysarthria, word retrieval deficits and strategy use. These relationships also make it impossible to state that mixed source repair is always more difficult to perform, or to assume that people with mild speech or mild word finding deficits always have less repairs. The clinical implication of this study is therefore that a combination of a motor speech disorder and different forms of linguistic impairment need to be considered in the clinical management of people with PD. Similarly, these factors should be considered in research on this group. Further, as in aphasia, training programmes are needed to help significant others and the healthcare staff who work with people with PD to learn how to facilitate the communication [48].

This study is based on limited data and uses a qualitative case study method. It includes only five cases with self-reported communication problems and the results cannot therefore be readily generalized to the population of those with PD as a whole. Nor can we be sure that the results we witnessed were not affected by age. Nevertheless, this field of research is still in its infancy. The findings from this study may provide a fruitful base for future research on larger, more representative samples where, for example, the relation between word retrieval capacity, degree of dysarthria and pragmatic skills may be further explored as well as the effect of the CP’s behavior in cases of communication problems.

## 5. Conclusions

In people diagnosed with PD, their motor speech disorder may be combined with word retrieval difficulties and communication strategies in ways that result in pragmatically atypical utterances. These may be difficult for communication partners to understand. People with PD and communication disorders are often dependent on their spouses to engage in cooperative repair and to re-establish mutual understanding. The compounding effects of motor speech disorder and language impairment need to be considered in the clinical management of and research on people with PD. Further research is required to see how common this type of speech and language complex is. However, this study adds to the growing evidence for the complexity of communication problems in PD.

## Figures and Tables

**Figure 1 brainsci-07-00123-f001:** Impaired speech, problem quickly resolved.

**Figure 2 brainsci-07-00123-f002:** Impaired speech, multiple attempts to repair.

**Figure 3 brainsci-07-00123-f003:** Word retrieval problem, quickly resolved.

**Figure 4 brainsci-07-00123-f004:** Word retrieval problem, prolonged repair sequence.

**Figure 5 brainsci-07-00123-f005:** Problematic topic shift.

**Table 1 brainsci-07-00123-t001:** Participant information and results on language and cognitive screening tests.

	Dyad 1		Dyad 2		Dyad 3		Dyad 4		Dyad 5	
	PwPD ^1^	CP ^2^	PwPD	CP	PwPD	CP	PwPD	CP	PwPD	CP
**Age**	78	73	67	61	78	76	70	66	63	55
**Education (years)**	11	10	*	*	9	7	7.5	10.5	8	13
**Years together**	51		42		57		41		18	
**Stage of PD ^3^**	V		IV		III		IV		IV	
**Comprehensibility in contextual speech**	39%		31%		83%		31%		23%	
**Token test (max: 261)**	*		259		193		169		228	
**Phonological verbal fluency (F-A-S)**	2-0-0		9-3-11		2-3-8		5-4-3		3-4-4	
**Semantic verbal fluency (animals-verbs)**	0-0		15-10		6-3		15-5		2-1	
**MMSE (max 30 ^4^)**	*		26 ^5^		22		*		29	

Notes: ^1^ = Person with Parkinson’s disease; ^2^ = Communication partner; * = Missing information; ^3^ = Stages of PD according to Hoehn & Yahr [37] focus on the movement disorder and runs from stage I (“Unilateral involvement only, usually with minimal or no functional impairment”) to stage V (“Confinement to bed or wheelchair unless aided”). It is thus not directly related to degree of communication disorder. ^4^ = Results below 24 points on the Mini Mental State Examination (MMSE) indicate cognitive decline. ^5^ = One visuospatial item in MMSE was not carried out by this participant due to the movement disorder.

**Table 2 brainsci-07-00123-t002:** Distribution of number of occurrences of repair in the five dyads.

	Dyad 1	Dyad 2	Dyad 3	Dyad 4	Dyad 5
**Amount of data in number of minutes**	30	30	30	60	105
**Number of occurrences of repair in each dyad (total: 221)**	26	21	14	31	129
**Number of occurrences of repair per minute**	0.9	0.7	0.5	0.5	1.2

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
