# Peer review of "Multifaceted Communication Problems in Everyday Conversations Involving People with Parkinson’s Disease"

_brainsci, 2017, doi:10.3390/brainsci7100123_

Round 1

Reviewer 1 Report

This article addresses an important gap in our understanding of the communication deficits associated with Parkinson’s disease.  The interaction among various speech, language and cognitive deficits has not been thoroughly investigated. Too often research is focused only on the motor aspects of communication, ignoring other important contributors to deficits. Qualitative methods, such as conversational analysis, offer tools appropriate for this topic. The manuscript is well-written and methodologically sound.  The following are some minor revisions that I hope will strengthen the manuscript.  I’ll present them roughly in chronological order of appearance.

·         In the introduction, consider adding a recent qualitative investigation of the experiences of people with PD.  Analysis of interviews adds to the growing literature suggesting the high cognitive demands of communication in PD.

Yorkston, K., Baylor, C., & Britton, D. (2017). Speech versus Speaking: The experiences of people with Parkinson’s disease and implications for intervention. American Journal of Speech-Language Pathology, 26, 561-568.

·         The author repeatedly use the term, “suffer” (lines70, 82, and 89) when describing the experiences of people with PD.  This term carries a somewhat negative connotation.  Consider using phrases such as “participants experience a motor speech disorder.”

·         In Table 1 references to the measurement tools should be provided along with an indication of the range of scores and the direction of better function.

·         On line 231, the word “what” is repeated.

·         On line 292, should SP-2 be CP-2?

·         In the Discussion around line 314, simplified syntax has also been reported in dysarthria associated with amyotrophic lateral sclerosis (Wilkinson, C., Yorkston, K. M., Strand, E., & Rogers, M. (1995). Features of spontaneous language in speakers with amyotrophic lateral sclerosis and dysarthria. American Journal of Speech-Language Pathology, 4(4), 139-142.)

Author Response

Response to Reviewer 1’s comments

We would like to offer our sincere thanks to reviewer 1 for the very constructive comments and help to improve the manuscript. We have now responded to the issues raised and have made the changes requested. Major changes and added information is highlighted with “track changes” in the manuscript. Responses to comments and information of what we have changed are provided below:

1. Yorkston et al (2017) have been added in the introduction (lines 88-89), thanks for helping us find this publication!

2. We have removed the expression “suffer” in the meaning of have/has PD all throughout the manuscript and replaced it with more suitable expressions (“has PD”, “diagnosed with PD” etc.)

3. In table 1 references to the measurement tools have been provided along with description of range of score (except for the fluency tasks where there are no set range) as well as information about the direction of better function. We are now also describing how the degree of comprehensibility was established as well as measures of language ability in the method section, lines 96-100 in this version of the manuscript.

4. The repeated what on former line 231 has been removed (thanks for spotting this)!

5. On former line 292 SP-2 has been amended to CP-2 (sorry about this remains from the Swedish language).

6. Thanks for the suggestion of the Wilkinson et al study (1995) regarding simplified syntax in ALS as a possible effect of the effort to speech. The reference has now also been included. 

Reviewer 2 Report

The authors present a qualitative analysis of communication between five persons with Parkinson’s disease (PwPD) and their spouses. This is a very-well written paper on a timely and important topic. More research is needed on the communication difficulties of persons with Parkinson’s disease, and reports of their impacts in real-life interaction are needed. My main concern is that the synergistic effect announced in the title is not strongly demonstrated in the results. Results should be reorganized and clarified, and some important discussion points should be added before this paper can be recommended for publication.

Lines 27-28. Dysarthria: A prevalence of up to 90% has been reported elsewhere (see for example Brabenec et al. 2017 for a recent review)

Lines 30-31. Voice problems, hypophonia: Do the authors mean “voice problems such as hypophonia” or “voice problems, i.e. hypophonia”? This should be clarified

Lines 33-42. The authors may want to cite Auclair-Ouellet, Lieberman and Monchi (in press) for a recent review.

Table 1.

The way the information for persons with PD and partners is reported (with a slash) is a bit confusing. Separate each member of the dyad in different columns.

Define the MMT acronym in table captions.

One patient is at stage V of Hoehn and Yahr. The link between the global clinical profile and the communication deficits should be put in perspective and discussed in more details.

Table 2.

It would be useful to have the number of repairs per minute to have a sense of the variability between dyads. Also, add individual speech comprehensibility scores.

Lines 147-148. Describe how the degree of comprehensibility was established. Later in the paper, it becomes clear that there was variability between PwPD in the degree of dysarthria. This information should be discussed here. Also, the presence (or absence) of a link between the degree of dysarthria and the number of repairs should be presented here.

Line 150. Report the number of repairs initiated because of speech in a more precise manner. For example, report results in terms of mixed speech/word-finding problems or speech/pragmatic problems, and so on. See also comment for line 326 below.

Lines 160-161. Major language comprehension deficits are not expected in PD. Was the speech comprehension deficit assessed more formally or was this impression only based on the interview? More details should be provided. On a related issue, were the PwPDs and CPs tested for hearing loss? If this is not the case, this limitation should be acknowledged. In general, the paper should discuss the partners’ communication style and attitudes, and the fact that CPs are also older adults who may have hearing loss and mild cognitive deficits.  

Line 168. Related to the previous comment: “CP-1 prioritises social interactions over the exchange of accurate information”. Does this apply to this specific instance only or it is characteristic of this partner’s communication style?  

Lines 158-192. It is mentioned towards the end of the discussion (lines 339-341) that dysarthric speech is hard to transcribe. This would contribute to the discussion of results presented here, showing that the need for repair cannot be attributed to limitations of the partner.

Line 220. Report comprehensibility score for this PwPD.

Line 239. On line 220, this PwPD speech disorder is qualified as “moderate”. However, on line 239, “His speech is severely dysarthric here”. Comment on the variability.

Line 318-319. The paper would make a stronger case for the importance of cooperation and support from the partner if it included a discussion of partner’s communication styles and attitudes (for example, Carlsson et al. 2014). Although they report research conducted in communication difficulties following traumatic brain injury, the papers of Ylvisaker et al. may provide interesting insights.

Line 326. The synergistic effect announced in the title and restated here is not strongly demonstrated by the data. This could be remediated by a more detailed presentation of the results, and an emphasis on synergy throughout the paper. However, due to the qualitative nature of analysis and the small number of participants, I feel it would be more appropriate to talk about “multiple communication problems” since the synergistic effect is hard to demonstrate. Nevertheless, answering the following question would be interesting:  Are mixed-source repairs harder to repair than repairs caused by only one source of problem? Do patients with mild speech or mild word finding deficits have less repairs? What is the role of the CP in that synergy? In the future, quantitative measures of word-finding capacity and pragmatic skills would be an important complement to the qualitative data. A more thorough description of communication partners would also be important to understand the dynamic within dyads.

Lines 336-337. This information should be presented earlier.

Lines 330-332. The authors mention that the spouse tended to do most of the talking. This is also an opportunity to comment on communication styles and their effects on interactions (Carlsson et al. 2014). 

Author Response

Response to Reviewer 2’s comments

We would like to offer our sincere thanks to reviewer 2 for the comments which have all been very helpful in improving the manuscript. We have responded to the issues raised and have made, as far as possible, the changes requested. Major changes and added information is highlighted with “track changes” in the manuscript. Responses to comments and information of what we have changed are provided below:

1. Lines 27-28: Regarding reference for prevalence of dysarthria in PD, we have changed reference to Ho et al 1999, (thanks for drawing our attention to this). However, based on the Ho et al paper we now say “Clinical-perceptual analysis have revealed that up to 89% of people diagnosed with Parkinson´s disease (PD) also have a progressive motor speech disorder, usually a hypokinetic dysarthria...” (as the number varies depending on what type of analysis and instruments used).

2. Lines 30-31: We have now clarified that we mean “voice problems, such as hypophonia”.

3. Lines 33-42: Thank you very much for the very interesting Auclair-Ouellet et al reference, it has now been included in the introduction.

4. Regarding Table 1: a) The ages and educational levels of the communication partners are now presented in a separate columns: b) The MMT acronym is now amended to MMSE and defined in the table caption; c) The lack of direct relation link between the global clinical profile in Hoenh & Yare stage and the communication deficits is now acknowledged in a table note.

5. Regarding Table 2:  Information of number of repairs per minute is now included, but not the individual speech comprehensibility scores, as those are not regarded as results and are instead provided in table 1.

6. Former Lines 147-148 (now lines 164-168): We are now discussing the (non direct) link between comprehensibility and occurrence of repair here. Further, we have added information on how the degree of comprehensibility was established in the method section, lines 98-100 in this version of the manuscript.

7. Response to comment regarding former line 150: Although we acknowledge the value of a reliable measure of distribution of repairs initiated of different reasons (for example, mixed speech/word-finding problems or speech/pragmatic problems), this is not possible to do as the different trouble sources are not mutually exclusive categories but different aspects which are present simultaneously in most of the repair sequences. We have now tried to explain why such account is not possible more thoroughly in lines 148-155.

8. Response to comment regarding former lines 160-161: Although no formal testing of hearing was performed all participant’s and their spouses were asked about their hearing (and the hearing of the PwPD), and impaired hearing that was not possible to compensate for by, for example, an hearing aid, was an exclusion criteria (this information is provided in the method section). PwPD-1 had an obvious cognitive decline which also affected his language comprehension and ability to participate in formal testing.  However, we do acknowledge that these questions raised here would probably also be raised by many other readers and we have now provided more information about this on lines 174-176. The issue about hearing is also important and we have now added a paragraph about this as in the discussion section, lines 379-384.

9. Response comment regarding former line 168:  This was characteristic for this CP’s communication style in all recordings, we have now clarified this in the text.

10. Comment regarding former lines 158-192: yes, thank you for pointing that out, we are now commenting on this in the amended discussion, where we now also discuss the role of the CPs. 

11. Comments regarding former line 220 and 239: Degree of comprehensibility is now reported here too for PwPD-4, we acknowledge that it adds important information about his communication. The classification of his degree of dysarthria as moderate has been removed as it refers to a result from a formal dysartria test and does not really mirror how functional his speech is in conversation.

12. Response to comments regarding former lines 318-319 and 330-332: Yes we agree and a discussion of the CPs role n the interaction has now been included in lines: 345-353 and 370-391.

13. Response comments regarding former line 326: We do acknowledge that we are not able to demonstrate that the effects seen in this data actually are synergistic effects. We have amended the title to “Multifaceted communication problems in everyday conversations. Further, we have now elaborated the discussion of CPs communication style and the role of the CP for the communication more and the role of the CP is now more included through the discussion but in particular discussed in lines 345-353 and 370-391.  Further, we do agree that the questions proposed by rev 2 are very interesting indeed, however, we feel that we are not able to provide any valid answers from this limited data set and instead we mention this and why we cannot provide simple answers lines 389-393 and also conclude in the discussion as issues for further research, a need for quantitative measures of word-finding capacity and pragmatic skill etc. 

14. Comment regarding former lines 336-337: The information about spouses’ tendency to refrain from initiating repair is not moved as it is related to the discussion about occurrences of word-retrieval problems. However it is now more integrated in the elaborated discussion of the role of the communication partner.

Round 2

Reviewer 2 Report

This paper represents an interesting contribution to the literature. The authors have done a great job at addressing the comments. My only remaining comments concern word choice and formulation.

Lines 377-378. The sentence starting by “At least” is a bit odd. I suggest making it the continuation of the previous sentence: This means that […] are always functional, at least not if […].

Line 378. Transference = transmission

Line 380. Higher age = older age

Line 381. May of course have impacts […]

Line 390-392. […] between factors such as degree of dysarthria […]

Line 392. These relationships also makes it […]

Author Response

Response to reviewer 2

Thank you for your positive comments! We are also grateful for your help with spotting the grammatical mistakes and your suggestions for revision of the wording. We have amended in accordance with all your suggestions.

Brain Sci. EISSN 2076-3425 Published by MDPI AG, Basel, Switzerland RSS E-Mail Table of Contents Alert
Back to Top